# Towards Effective Multi-Modal Interchanges in Zero-Resource Sounding Object Localization

**Yang Zhao**[*], **Chen Zhang**[*], **Haifeng Huang**[*], **Haoyuan Li, and Zhou Zhao** [†]

Zhejiang University

Alibaba-Zhejiang University Joint Research Institute of Frontier Technologies

`{awalk, zc99, huanghaifeng, lihaoyuan, zhaozhou}@zju.edu.cn`

## Abstract

Aiming to locate the object that emits a specified sound in complex scenes, the task of sounding object localization bridges two perception-oriented modalities of vision and acoustics, and brings enormous research value to the comprehensive perceptual understanding of machine intelligence. Although there are massive training data collected in this field, few of them contain accurate bounding box annotations, hindering the learning process and further application of proposed models. In order to address this problem, we try to explore an effective multi-modal knowledge transfer strategy to obtain precise knowledge from other similar tasks and transfer it through well-aligned multi-modal data to deal with this task in a zero-resource manner. Concretely, we design and propose a novel *Two-stream Universal Referring localization Network* (TURN), which is composed of a localization stream and an alignment stream to carry out different functions. The former is utilized to extract the knowledge related to referring object localization from the image grounding task, while the latter is devised to learn a universal semantic space shared between texts and audios. Moreover, we further develop an adaptive sampling strategy to automatically identify the overlap between different data domains, thus boosting the performance and stability of our model. The extensive experiments on various publicly-available benchmarks demonstrate that TURN can achieve competitive performance compared with the state-of-the-art approaches without using any data in this field, which verifies the feasibility of our proposed mechanisms and strategies. The code is available at https://github.com/AwalkZY/TURN.

## 1 Introduction

When interacting with the external environment, humans can gather and integrate much information through different types of perception (such as vision, hearing, smell, and taste), thus making corresponding judgments and decisions based on this. To effectively mimic this ability to bridge knowledge from various modalities, researchers have been trying to explore the potential of machine intelligence from different aspects. Among them, as a crucial part of audio-visual learning, the task of sounding object localization has gained an increasing amount of attention.

Given a specific sound, the goal of sounding object localization is to ascertain the object that emits it in a complex scene. To tackle this problem, a lot of efforts are made to model the co-occurrence patterns in the visual and acoustic data and accordingly identify the possible target region. And for the research and development in this field, there have been numerous aligned data pairs collected into several publicly-available benchmarks. Nevertheless, few of them contain available box annotations, which forces the models to acquire knowledge in a weakly supervised manner and increases the

---

[*]Equal Contribution.

[†]Corresponding Author.

36th Conference on Neural Information Processing Systems (NeurIPS 2022).

difficulty in the training process. Apart from this, the annotations in the test split are usually given as bounding boxes, which keeps consistent with other spatial localization tasks. But restricted by the architecture and learning paradigm, most of the proposed models can only generate results in the format of heatmaps but not bounding boxes, bringing much inconvenience to the further utilization of located regions.

In spite of this, when we look into another subproblem of referring object localization, i.e., image grounding, we may find that the data in this field are usually equipped with accurate bounding box annotations, which brings higher information density and better model performances. So is there any suitable solution to extract and migrate the knowledge from this field to our target scenario? The answer is **Yes**. As with the pivot-based alignment strategies adopted in the zero-resource machine translation [61, 40, 23, 52], we can turn to the datasets of audio retrieval and use a series of aligned text-audio pairs as intermediaries to establish an efficient knowledge transfer pipeline to tackle the target problem in a zero-resource manner. Although it sounds straightforward and feasible theoretically, such a learning scheme is still faced with two kinds of practical challenges: 1) ***Domain gap***, which consists of the essential discrepancies between all these three data domains. As depicted in the green part of Figure 1, the queries in image grounding put more emphasis on the *appearance, quantity and spatial relationship* of the objects, while those in audio retrieval tend to describe *action, state and temporal change* about the targets. Meanwhile, similar gaps also exist in the visual inputs (illustrated in the red part) and the acoustic data in the other two modality pairs. 2) ***Modality gap***, which stems from the intrinsic properties within different modalities and the complicated relationships between them. The noise and redundant information in these three modalities will hamper the transfer process from different aspects, and the non-monotonic alignment between modalities also makes it hard to learn cross-modal representation in fine granularity.

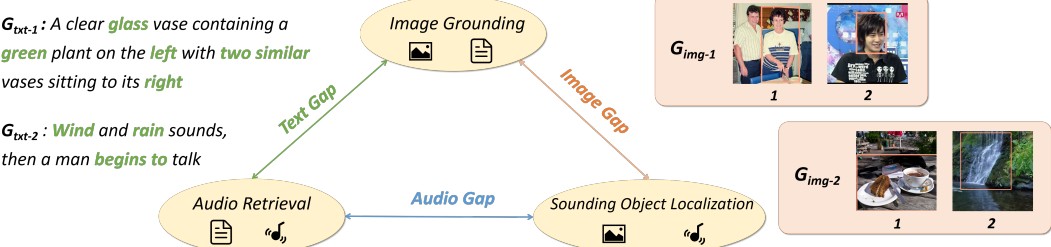

Figure 1: The major challenges in our problem setting.

To this end, we propose a novel *Two-stream Universal Referring localization Network* (TURN) to handle the problem of sounding object localization in a zero-resource way and reduce the potential performance degradation from the challenges described above. Specifically, we utilize a localization stream to model the relationship between visual elements and referring queries so as to unearth valuable location knowledge from image grounding data. Meanwhile, we develop an alignment stream to construct a universal semantic space for the linguistic and acoustic data, thus enabling efficient knowledge transfer to our target task. Moreover, considering that different data items will affect the transferability of knowledge to a varying extent, we further devise an adaptive sampling strategy to automatically identify the overlap between various data domains and accordingly modulate the sampling weights of data, which enhances the stability of the learning process and improves the all-around performance of our model.

In conclusion, our contributions in this paper can be summarized in the following aspects:

- We explore the task of sounding object localization from a zero-resource perspective for the first time and discuss the feasibility of transfer-based solutions in this setting.

- We devise a novel adaptive sampling mechanism in the area of multi-modal knowledge transfer, which can automatically discover the overlap between different data domains and adjust the sampling weights of data, thus enhancing the comprehensive performance of the learning model.

- We propose an effective two-stream architecture for the target of zero-resource sounding object localization. Without using any training data in this field, TURN achieves competitive

performance on publicly-available benchmarks, including MUSIC and VGGSS, which demonstrates the feasibility of our proposed solution.

## 2 Related Work

**Audio-Text Representation Learning**   Cross-modal representation learning aims to capture the relationship between different modalities and project embeddings to the same latent space. Though many works have studied image-text [22, 15, 44] and audio-visual representation learning [5, 38, 53], audio-text representation learning is still under-studied. Some works [38, 8, 35] directly computed the similarity between embeddings but cannot well encode the semantic and acoustic information into the embeddings at the same time. Other works such as Chen et al. [12] attempted to learn interactive cross-modal representation with both acoustic and textual information followed the spirit of contrastive language–image pre-training, but might be deeply affected by the insufficiency of training data. Considering that VQ-VAE [56, 6, 68] has shown its outstanding performance in discrete representation learning, we try to learn the audio-text representation via cross-modal reconstruction and codebook sharing based on the structure of VQ-VAE.

**Transfer Learning**   Typical transfer learning is usually applied to solve the problem of data distribution inconsistency between source and target domains. The mainstream approaches in this area usually tackle this problem from one of these three perspectives: data inputs [16, 3, 66, 11], intermediate representations [9, 41, 20, 31, 58, 67] and learning paradigms [36, 30, 32, 18, 46]. However, these works are based on the assumption that the source and target domains are conceptually close and modally consistent. In order to handle the situation where data from the source domain and target domain are not closely related, Tan et al. [54] formulated the paradigm of *Transitive Transfer Learning* (TTL) and exploited an auxiliary annotated intermediate domain to transfer knowledge. And considering the cost of constructing the intermediate domain, Tan et al. [55] further proposed a novel learning schema *Distant Domain Transfer Learning* (DDTL), in which the intermediate domain will be automatically selected and constructed from multiple datasets without labels, thus achieving the knowledge transfer between completely unrelated domains. Recently, some researchers have also discussed the usage of transfer learning strategies in the multi-modal scenarios, which covers the field of image caption [39], audio caption [71], machine translation [40] and so forth. In this paper, we further explore the application of transfer learning paradigms in the task of zero-resource sounding object localization as a complement to this field.

**Referring Object Localization**   Given a complicated scene with multiple objects and a specific referential query, the task of referring object localization is to ascertain the location of the object that best matches the given reference. Depending on the modality of queries, this field can be subdivided into *Image Grounding* and *Sounding Object Localization*. In the scenario of image grounding, the queries are given in natural language descriptions, and fully-annotated data are denoted to facilitate this area. Consequently, a lot of supervised approaches and architectures are proposed to tackle this problem, which can be further categorized into *proposal-based* [24, 28, 34, 59, 60, 62, 69, 72] and *proposal-free* ones [17, 33, 47, 64, 63]. The former estimate the confidences of all the pre-defined proposals and choose the best one in a ranking-based way, while the latter directly regress the coordinates of bounding boxes based on the fully aggregated and fused representations. When it comes to acoustic queries, clear box annotations are usually unavailable in the training partition of previously proposed datasets [14, 48, 70]. Therefore, researchers in this field turn to explore the solutions [26, 42, 45, 51, 1, 25] based on contrastive learning paradigms and produce pixel-level heatmap results in a weakly-supervised way. Although *sounding object localization* shares lots of commonalities in essence with *image grounding*, there is still a lack of discussions and analyses of their relationships and characteristics. In this paper, we make a pioneering attempt to link these two subproblems and explore modality-independent universal referring object localization.

## 3 Method

### 3.1 Problem Formulation

As depicted in Figure 2, our goal is to extract adequate knowledge from the data of image grounding and modulate it with the assistance of well-aligned audio-text corpora in the field of audio retrieval,

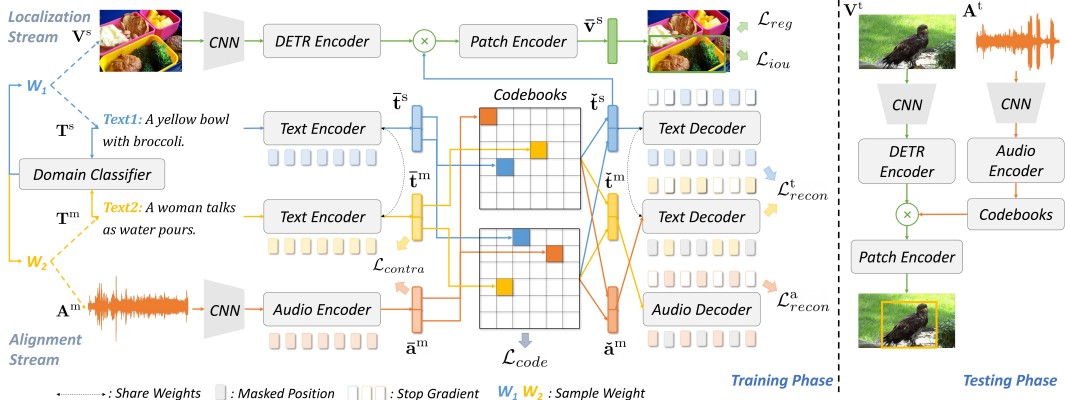

Figure 2: Overall architecture diagram of our model *TURN*. Best viewed in color mode and with zoom-in. The green line is *localization* stream and other lines are combined to form *alignment* stream.

thus achieving an acceptable generalization performance on the task of sounding object localization. For the sake of clarity, we regard the data of image grounding, audio retrieval and sounding object localization are from the *source*, *intermediate* and *target* domains with the notations of $\mathbf{D}^s$, $\mathbf{D}^m$ and $\mathbf{D}^t$, respectively. Under this definition, this problem setting can be also considered as a knowledge transfer and generalization from $\mathbf{D}^s$ to $\mathbf{D}^t$ via $\mathbf{D}^m$. To keep consistent with the other prior works and the characteristics of different domains, only a limited amount of data from *source* and *intermediate* domains are accessible to the model during training, and the annotated testing data of *target* domain will be finally used to measure the overall generalization ability in the evaluation.

## 3.2 Model Architecture and Data Specification

**Overall Network Structure**     The overall workflow of our proposed model is depicted in Figure 2. Specifically, our model conducts the calculations in two parallel pathways referred to as *localization* and *alignment* streams in the following descriptions. The *localization* stream focuses on modeling dependencies within visual elements and capturing relationships between inputs and queries, thus accordingly producing the target bounding boxes. In this part, a series of pre-trained backbones are first utilized to embed the input images into a group of object-aware features. And these features will be further refined and aggregated with the guidance of query conditions and then used to predict the coordinates of target regions. Meanwhile, the *alignment* stream is developed to align the data distribution of different domains and modalities into a shared space and generate debiased query conditions to serve as references for the *localization* stream. In order to construct a semantically complete latent space of query embeddings, we exploit a group of shared vector quantizers and devise a cross-reconstruction mechanism to alleviate the domain and modality gaps existing in the input audio and text data. Moreover, we additionally employ an auxiliary classifier to adjust the sampling weight during the training phase to help our model adaptively absorb knowledge from different domains.

**Inputs and Annotations**     Before the detailed descriptions of our proposed architecture, we first formulate the concrete symbolic definitions of the inputs, annotations and representations. To begin with, the original data from *source*, *intermediate* and *target* domains are denoted as $\mathbf{D}^s = \{(\mathbf{V}^s, \mathbf{T}^s, \mathbf{b}^s) \in (\mathcal{V}^s, \mathcal{T}^s, \mathcal{B})\}$, $\mathbf{D}^m = \{(\mathbf{T}^m, \mathbf{A}^m) \in (\mathcal{V}^m, \mathcal{T}^m)\}$ and $\mathbf{D}^t = \{(\mathbf{V}^t, \mathbf{A}^t, \mathbf{b}^t) \in (\mathcal{V}^t, \mathcal{T}^t, \mathcal{B})\}$. For simplification, we omit the superscript in the following explanations when there is no need to distinguish between different domains. In these notations, $\mathbf{V} \in [0, 1]^{3 \times H \times W}$, $\mathbf{T} \in \mathbb{N}^{L_t}$ and $\mathbf{A} \in \mathbb{R}^{L_m \times c}$ represent the pixel matrix, word index sequence, and log-mel spectrogram of the image, text and audio inputs, respectively. As for the annotation format, $\mathbf{b} = [x, y, w, h] \in [0, 1]^4$ represents the target bounding box of queried object.

## 3.3 Localization Stream

As the basic component of referring object localization, the *localization* stream is applied to obtain location-related knowledge from the source domain and produce target bounding box predictions.

Given an image containing multiple objects $\mathbf{V} \in [0,1]^{3 \times H \times W}$ and a semantically condensed query condition $\mathbf{q} \in \mathbb{R}^d$, we first project the visual input into a series of object-aware embeddings $\tilde{\mathbf{V}} = \{\tilde{\mathbf{v}}_1, \ldots, \tilde{\mathbf{v}}_{n_v}\}$ via a pre-trained DETR backbone [10] and a group of multi-layer perceptrons, where $\tilde{\mathbf{v}} \in \mathbb{R}^d$ and $n_v$ is the input sequence length of the DETR encoder.

After the initial extraction procedure, we combine the guidance information from query conditions into the visual representations via a simple element-wise multiplication. And then, we employ another transformer encoder component to model the dependencies and relations among these query-aware visual features and generate a bunch of fully fused representations, which can be given by

$$\bar{\mathbf{v}} = \tfrac{1}{n_v} \sum(\Theta_{\mathrm{v}}(\hat{\mathbf{v}}_1, \hat{\mathbf{v}}_2, \ldots, \hat{\mathbf{v}}_{n_v})), \quad \text{where} \ \ \hat{\mathbf{v}}_i = \tilde{\mathbf{v}}_i \cdot \mathbf{q} / \|\tilde{\mathbf{v}}_i \cdot \mathbf{q}\|_2. \tag{1}$$

In this formula, the notation $\Theta(\cdots)$ stands for the transformer encoder module and we conduct a $l_2$-norm to prevent numerical instability caused by the dot product calculation.

Finally, we calculate the aggregated representation by averaging all the patch features and utilize a multi-layer perceptron to generate the prediction of bounding boxes, given by

$$\hat{\mathbf{b}} = \sigma(\mathbf{W}_2^{\mathrm{v}}(\mathrm{ReLU}(\mathbf{W}_1^{\mathrm{v}} \bar{\mathbf{v}} + \mathbf{c}_1^{\mathrm{v}})) + \mathbf{c}_2^{\mathrm{v}}), \tag{2}$$

where $\mathbf{W}_1^{\mathrm{v}} \in \mathbb{R}^{d \times d}, \mathbf{W}_2^{\mathrm{v}} \in \mathbb{R}^{4 \times d}$ and $\mathbf{c}_1^{\mathrm{v}}, \mathbf{c}_2^{\mathrm{v}} \in \mathbb{R}^d$ are all learnable parameters, $\sigma(\cdot)$ is the sigmoid function and $\hat{\mathbf{b}} \in [0,1]^4$ is the predicted bounding box.

## 3.4 Alignment Stream

In the overall knowledge transfer procedure, the alignment stream plays a crucial role in extracting the domain- and modality-invariant representations from the original inputs and providing query condition features as the reference of the localization stream.

In the training phase, we first project the text data into a series of continuous word embeddings, denoted as $\hat{\mathbf{T}}^{\mathrm{s}} = \{\hat{\mathbf{t}}_1^{\mathrm{s}}, \ldots \hat{\mathbf{t}}_{L_t}^{\mathrm{s}}\}$ and $\hat{\mathbf{T}}^{\mathrm{m}} = \{\hat{\mathbf{t}}_1^{\mathrm{m}}, \ldots, \hat{\mathbf{t}}_{L_t}^{\mathrm{m}}\}$, respectively. And the log-mel spectrogram of audio data will be compressed via a stack of convolutional layers, given as $\hat{\mathbf{A}}^{\mathrm{m}} = \{\hat{\mathbf{a}}_1^{\mathrm{m}}, \ldots, \hat{\mathbf{a}}_{L_a}^{\mathrm{m}}\}$. Afterward, we utilize two transformer encoders to generate context-aware linguistic and acoustic features and aggregate them into a single semantically-rich representation $\bar{\mathbf{t}}, \bar{\mathbf{a}} \in \mathbb{R}^d$, as formulated in Equation 3. It's worth noting that the word embedding table and text encoder are shared between source and intermediate domains so as to filter out the domain-specific bias within the original data.

$$\bar{\mathbf{t}} = \tfrac{1}{L_t} \sum(\Theta_{\mathrm{t}}(\hat{\mathbf{t}}_1, \ldots, \hat{\mathbf{t}}_{L_t})), \quad \bar{\mathbf{a}} = \tfrac{1}{L_a} \sum(\Theta_{\mathrm{a}}(\hat{\mathbf{a}}_1, \ldots, \hat{\mathbf{a}}_{L_a})), \tag{3}$$

Next, we expect to learn a universal semantic representation space where the data with the same semantics but from different domains or modalities will be mapped to adjacent positions so that the localization of target objects can be independent of the query source. Regarding the remarkable performance of variational autoencoder in fitting distributions of variables, we employ the straight-through VQ-VAE [56] module to achieve this goal. Following the strategy adopted in [7], we split the aggregated features into a total of $q$ chunks and utilize a group of codebooks $\{\Psi_1(\cdot), \ldots, \Psi_q(\cdot)\}$ to project them into multiple shared semantic spaces and reorganize them into a series of debiased query representations $\check{\mathbf{t}}, \check{\mathbf{a}} \in \mathbb{R}^d$, which will serve as the query condition term $\mathbf{q}$ in Equation 1 and can be given by

$$\check{\mathbf{t}} = [\Psi_1(\bar{\mathbf{t}}_1); \cdots; \Psi_q(\bar{\mathbf{t}}_q)], \quad \check{\mathbf{a}} = [\Psi_1(\bar{\mathbf{a}}_1); \cdots; \Psi_q(\bar{\mathbf{a}}_q)], \tag{4}$$

where $[\,;]$ is the concatenation operator and $\bar{\mathbf{t}}_i, \bar{\mathbf{a}}_i \in \mathbb{R}^{d/q}$ represents the $i$-th part of $\bar{\mathbf{t}}$ and $\bar{\mathbf{a}}$ (i.e., $\bar{\mathbf{t}} = [\bar{\mathbf{t}}_1; \ldots; \bar{\mathbf{t}}_q]$ and $\bar{\mathbf{a}} = [\bar{\mathbf{a}}_1; \ldots; \bar{\mathbf{a}}_q]$)

Afterward, we further devise a *self-cross-reconstruction* mechanism to improve the robustness of query representations and enhance the mutual information between aligned texts and audios.

$$\tilde{\mathbf{T}}^{\mathrm{s}} = \Omega_{\mathrm{t}}(\check{\mathbf{t}}^{\mathrm{s}}, \mathbf{T}_{mask}^{\mathrm{s}}), \quad \tilde{\mathbf{T}}^{\mathrm{m}} = \Omega_{\mathrm{t}}(\check{\mathbf{a}}^{\mathrm{m}}, \mathbf{T}_{mask}^{\mathrm{m}}), \quad \tilde{\mathbf{A}}^{\mathrm{s}} = \Omega_{\mathrm{a}}(\check{\mathbf{t}}^{\mathrm{m}}, \mathbf{A}_{mask}^{\mathrm{m}}), \tag{5}$$

where $\mathbf{T}_{mask}$ and $\mathbf{A}_{mask}$ are the modified inputs in which a certain proportion of the original data into the *<mask>* embedding and $\Omega(\mathbf{X}, \mathbf{Y})$ is a mask-predict transformer decoder to reconstruct the masked elements of $\mathbf{Y}$ with the reference of $\mathbf{X}$. As formulated in Equation 5, the texts from the source domain are reconstructed with the guidance of itself and the reconstruction in the intermediate domain will be performed in a cross-modality way. Thereafter, the aggregated features of one modality will be able to get modulated in a data-driven manner to serve as reasonable substitutes of the other one.

### 3.5 Training and Inference

As mentioned above, the entire architecture can be trained in an end-to-end mechanism. Given the annotation $\mathbf{b}$ from the source domain, we follow the learning strategy adopted in [17] and apply a combination of regression-based and IoU-based loss functions to optimize the *localization* stream, formulated as

$$\mathcal{L}_{loc} = \lambda_1 \mathcal{L}_{reg}(\hat{\mathbf{b}}, \mathbf{b}) + \lambda_2 \mathcal{L}_{iou}(\hat{\mathbf{b}}, \mathbf{b}) \tag{6}$$

where the $\mathcal{L}_{reg}(\cdot, \cdot)$ is the smooth $l_1$ loss to optimize the regressions of each coordinate separately, and $\mathcal{L}_{iou}(\cdot, \cdot)$ is the GIoU loss to conduct an overall constraint on predicted boxes.

As for the *alignment* stream, the optimization target is composed of the following three aspects, which will be presented in the logical order of calculations. Initially, in order to maximize the possibility of mapping the aligned linguistic and acoustic data to the same discrete code, we apply a contrastive learning paradigm with the InfoNCE loss [50, 57] in every mini-batch to constrain the distribution of aggregation vectors, given by

$$\mathcal{L}_{contra} = \frac{1}{b} \sum_{i=1}^{b} \log\left(\frac{\exp(\mathcal{D}(\bar{\mathbf{t}}_i^{\mathrm{m}}, \bar{\mathbf{a}}_i^{\mathrm{m}})/\tau_c)}{\sum_{j=1}^{b} \exp(\mathcal{D}(\bar{\mathbf{t}}_i^{\mathrm{m}}, \bar{\mathbf{a}}_j^{\mathrm{m}})/\tau_c)}\right), \tag{7}$$

where $b$ is the size of each mini-batch and $\tau_c$ is the temperature hyper-parameter. $\mathcal{D}(\mathbf{x}, \mathbf{y})$ measures the distance between two vectors $\mathbf{x}$ and $\mathbf{y}$, and we use the the normalized inner-product to calculate the cosine-similarity distance in this part.

And next, all the codebooks will conduct calculations in parallel and get optimized in a $K$-means based manner individually, which can be abstracted as

$$\mathcal{L}_{code} = ||\mathrm{sg}[\mathbf{f}] - \Psi(\mathbf{f})||_2^2 + \beta||\mathbf{f} - \mathrm{sg}[\Psi(\mathbf{f})]||_2^2, \tag{8}$$

where $\mathbf{f}$ denotes the input of codebooks and $\mathrm{sg}(\cdot)$ represents the stop-gradient operation.

Moreover, two reconstruction losses will be utilized to indirectly modulate condition representations, given by

$$\mathcal{L}_{recon}^{\mathrm{t}} = \mathcal{L}_{ce}(\tilde{\mathbf{T}}, \mathbf{T}), \quad \mathcal{L}_{recon}^{\mathrm{a}} = \mathcal{L}_{l_1}(\tilde{\mathbf{A}}, \mathbf{A}) \tag{9}$$

where $\mathcal{L}_{ce}(\cdot, \cdot)$ and $\mathcal{L}_{l_1}(\cdot, \cdot)$ are the cross-entropy and $l_1$ loss functions, respectively.

In conclusion, the optimization of *alignment* stream is summarized as

$$\mathcal{L}_{aln} = \lambda_3 \mathcal{L}_{code} + \lambda_4(\mathcal{L}_{recon}^{\mathrm{a}} + \mathcal{L}_{recon}^{\mathrm{t}}) + \lambda_5 \mathcal{L}_{contra}, \tag{10}$$

Finally, the overall loss function is combined as $\mathcal{L} = \mathcal{L}_{loc} + \mathcal{L}_{aln}$ to learn our proposed architecture, where $\lambda_1, \ldots, \lambda_5$ are the balancing factors to control the magnitude of corresponding terms. And in the process of inference, we can simply take the projected embeddings of audio data as the condition representations to the localization stream and generate the calculation result.

### 3.6 Adaptive Sampling Strategy

Although we have already obtained a set of domain- and modality-invariant representations and performed sufficient alignments, the huge inherent gaps between the source and intermediate domains still heavily hamper the transferability and generalization ability. As illustrated previously, there tend to be some domain-specific data samples in the corpora. For example, *"The half of the sandwich on the right"* might be a good item for image grounding, but there merely exists any acoustic information in this scenario. And it will also be too abstract to imagine the concrete object corresponding to *"A loud high-pitched sound takes place."* Generally, a sample like *"A car was going on the road."* may serve as an excellent point to bridge knowledge. To this end, we try to estimate the *domain-exclusivity* of data items and take this as a reference for data sampling. Specifically, given all the queries from $\mathcal{T}^{\mathrm{s}}$ and $\mathcal{T}^{\mathrm{m}}$, we first employ an extra classifier $\Phi(\cdot)$ to discriminate between them and use the domain labels to optimize this module, formulated as

$$\mathcal{L}_{ext} = -\sum_{\mathbf{t}^{\mathrm{s}} \in \mathcal{T}^{\mathrm{s}}} \log(\sigma(\Phi(\mathbf{t}^{\mathrm{s}}))) - \sum_{\mathbf{t}^{\mathrm{m}} \in \mathcal{T}^{\mathrm{m}}} \log(1 - \sigma(\Phi(\mathbf{t}^{\mathrm{m}}))). \tag{11}$$

After this training procedure, the classifier will respond with higher confidence to the data from the source domain and lower confidence to those from the intermediate domain. On this basis, we can

safely assume that $\mathbf{t}^{\mathrm{s}}$ with **lower** confidence $\Phi(\mathbf{t}^{\mathrm{s}})$ and $\mathbf{t}^{\mathrm{m}}$ with **higher** $\Phi(\mathbf{t}^{\mathrm{m}})$ ought to be shared between these two domains and located within the intersection part.

Consequently, the data sampling weights will be assigned adaptively via the following mechanism.

$$w_i^{\mathrm{s}} = \frac{\exp(-\Phi(\mathbf{t}_i^{\mathrm{s}})/\tau^{\mathrm{s}})}{\sum_j \exp(-\Phi(\mathbf{t}_j^{\mathrm{s}})/\tau^{\mathrm{s}})}, \qquad w_i^{\mathrm{m}} = \frac{\exp(\Phi(\mathbf{t}_i^{\mathrm{m}})/\tau^{\mathrm{m}})}{\sum_j \exp(\Phi(\mathbf{t}_j^{\mathrm{m}})/\tau^{\mathrm{m}})}, \tag{12}$$

and we have

$$\tau^* = \frac{\max(\Phi(\mathbf{t}^*)) - \min(\Phi(\mathbf{t}^*))}{\ln(k)}, \tag{13}$$

where $w_i^*$ is the probability for the $i$-th item to be selected into mini-batches in the training phase. In this formula, the term $\tau^*$ is applied to smooth the distribution of sampling weights and $k$ is the expected ratio of the maximum and minimum in all weights, constraining the range of possible results and reducing the impacts of outliers to some extent.

# 4 Experiments and Results

## 4.1 Datasets and Metrics

**Image Grounding Datasets** We choose RefCOCO [65]/RefCOCO+ [65]/RefCOCOg [37] as the image grounding datasets and conduct in-depth studies on the RefCOCOg dataset. The statistic information of these three datasets can be found on the webpage of Tensorflow. [3] We follow the data partitions from Yu et al. [65][4] and only use the training set in the overall learning process.

**Audio Retrieval Datasets** As for the audio-text datasets, we use Clotho [19] and AudioCaps [29]. Clotho is sourced from the online platform Freesound [21] and contains 4,981 audio samples and 24,905 captions (5 captions for each audio clip). AudioCaps contains video clips and corresponding natural language captions, and we only use the audio clips extracted from the videos, resulting in totally 50,960 audio-text pairs for training.

**Sounding Object Localization Datasets** For the dev set, we use the annotated subset of Flickr-SoundNet [48], which is built on Flickr-SoundNet dataset [4] and contains 2,786 annotated image-audio pairs. For the test set, we choose VGGSS [14] and MUSIC [70]. VGGSS is derived from VGGSound [13] and contains 5,158 audio-image pairs. MUSIC, as a musical instrument video dataset, consists of 685 untrimmed videos of musical solos and duets spanning 11 instrument categories. We follow the data process pipeline of Hu et al. [27], and finally we use 489 solo videos for test.

**Metrics** Following [43, 48, 14], we adopt Consensus Intersection over Union (cIoU) and Area Under Curve (AUC) to evaluate the performance of our model. It's noteworthy that cIoU and vanilla IoU will be equivalent under the circumstance of $n = c = 1$ as shown in Equation 14.

$$\bar{\mathbf{m}} = \min\left(\sum_{j=1}^n \frac{\mathbf{m}_j}{c}, 1\right), \quad \mathrm{cIoU}(\{\mathbf{m}_j\}_{j=1}^n, \hat{\mathbf{m}}) = \mathrm{IoU}(\bar{\mathbf{m}}, \hat{\mathbf{m}}), \tag{14}$$

where $\{\mathbf{m}_j\}_{j=1}^n$ and $\hat{\mathbf{m}}$ are the binary masks converted from ground-truth and predicted boxes.

## 4.2 Comparisons with State-of-the-art Methods

Though TURN is designed for the zero-resource scenario, it still reaches a counterpart to other full-resource state-of-the-art sounding object localization methods. Here we compare our TURN with previous leading methods on VGGSS and MUSIC-Solo as Table 1 and Table 2 respectively. And it is noteworthy that we retain all the samples with multiple box annotations of VGGSS in the entire evaluation process. We can find that TURN outperforms all of the previous methods in VGGSS in both cIoU and AUC metrics. In MUSIC-Solo, the results are similar – TURN obtains the best AUC among all the methods with a reasonable IoU score. It is also worth noting that the cIoU and AUC of both datasets suffer a sharp drop when a random vector takes the place of the actual query in the localization stream (denoted as TURN (w/o Query)). This observation shows that it is unreliable to retrieve the salient region as the predictions by only the localization stream and further indicates

---

[3]https://www.tensorflow.org/datasets/catalog/ref_coco
[4]https://github.com/lichengunc/refer

the validity of our proposed knowledge transfer strategy. Apart from this, we can also note that the IoU@0.5 metric on the MUSIC-Solo dataset is relatively low when compared to other methods. We infer that this phenomenon may stem from the inherent annotation gap between image grounding and sounding object localization, and the low proportion of music instruments in the training datasets.

Table 1: Performances on VGGSS dataset.

| Method | cIoU@0.5 | AUC |
|---|---|---|
| Attention [48] | 17.1 | 28.7 |
| AVobject [1] | 29.7 | 35.7 |
| HardWay [14] | 31.9 | 37.0 |
| LessMore [49] | 32.2 | 36.6 |
| SSPL [51] | 33.9 | 38.0 |
| TURN (w/o Query) | 12.2 | 26.9 |
| TURN (w/ Query) | **34.6** | **39.1** |

Table 2: Performances on MUSIC-Solo dataset.

| Method | IoU@0.5 | AUC |
|---|---|---|
| Object-that-sound [2] | 26.1 | 35.8 |
| DMC [25] | 29.1 | 38.0 |
| Attention [48] | 37.2 | 38.7 |
| Sound-of-pixel [70] | 40.5 | 43.3 |
| DSOL [26] | **51.4** | 43.6 |
| TURN (w/o Query) | 16.9 | 34.1 |
| TURN (w/ Query) | 33.7 | **45.2** |

## 4.3 Ablation Studies

In this part, we conduct ablation studies to verify the effectiveness of each design detail in TURN according to the results on VGGSS and MUSIC-Solo.

**Choice of Source and Intermediate Datasets** TURN needs to generalize knowledge from the source domain to the target domain with the assistance of the intermediate domain. Therefore, we make a comparison between different selections of source and intermediate datasets, which is illustrated in Table 3. Consistent with our expectations, the best choice for the source domain is RefCOCOg. Compared with RefCOCO and RefCOCO+, the grounded textual descriptions in RefCOCOg contain more details about the object itself, which is vital when aligned to the texts from the intermediate domain. As for intermediate domain datasets, it is straightforward to find that AudioCaps is more suitable for the MUSIC-Solo dataset, and Clotho helps VGGSS more. The reason can be attributed to the fact that AudioCaps contains more instrument-related entries and indicates the names of different instruments in the descriptions, while the audio quality in Clotho is much higher, and there are multiple descriptions assigned to the same audio, which reduces noise disturbances in general.

Table 3: The results of different source and intermediate domains on VGGSS and MUSIC-Solo.

| Source | Intermediate | VGGSS | | MUSIC | |
|---|---|---|---|---|---|
| | | cIoU@0.5 | AUC | IoU@0.5 | AUC |
| RefCOCO | AudioCaps | 23.13 | 33.89 | 15.87 | 36.06 |
| RefCOCO+ | AudioCaps | 27.26 | 35.27 | 29.12 | 43.95 |
| RefCOCOg | AudioCaps | 33.58 | 38.57 | 33.68 | **45.17** |
| RefCOCO | Clotho | 16.40 | 31.50 | 21.43 | 41.68 |
| RefCOCO+ | Clotho | 22.51 | 33.19 | **36.47** | 42.89 |
| RefCOCOg | Clotho | **34.57** | **39.07** | 28.21 | 42.42 |

**Effect of Different Training Settings** In order to verify the effectiveness of regularization terms in our model, we conduct an individual ablation study on each term. As illustrated in Table 4, the contrastive loss takes the most significant effect, implying it plays a leading role in the alignment of data from different modalities. And the codebook loss optimizes the learning process of shared semantic space, determining the effect of knowledge transfer. Although the reconstruction loss has no direct influence on representations, it can still improve the performance by enhancing mutual information between different modalities. In addition, we also evaluate the adaptive sampling strategy, and the performance margin indicates the benefit it brings to the learning process.

**Choice of Hyper-parameters in Codebooks** As described in Section 3.4, we use a group of codebooks to project aggregated features in the alignment stream. Here we investigate how the group number of codebooks and total codebook size influence the localization performance. From Figure 3, we can observe that the total codebook size cannot be either too large (depicted by the green line, the embeddings from different modalities might be scattered into different partitions of the codebook space) or too small (shown by the blue line, the codebook capacity is not enough). And given a fixed

Table 4: The performance of different experimental settings on VGGSS dataset.

| Contrastive Loss | Codebook Loss | Reconstruction Loss | Adaptive Sampling | Clotho | | AudioCaps | |
|:---:|:---:|:---:|:---:|:---:|:---:|:---:|:---:|
| | | | | cIoU | AUC | cIoU | AUC |
| ✓ | ✓ | ✓ | ✓ | **34.57** | **39.07** | **33.58** | **38.57** |
| ✓ | ✓ | ✓ | | 28.56 | 36.75 | 29.24 | 36.55 |
| ✓ | ✓ | | ✓ | 30.09 | 36.55 | 28.42 | 36.09 |
| ✓ | | ✓ | ✓ | 27.53 | 36.37 | 27.70 | 36.13 |
| | ✓ | ✓ | ✓ | 26.44 | 34.62 | 21.21 | 34.23 |

total size, the codebook group number should also be tuned in a proper range. According to the experiment results, when the codebook number is set to be 2 and the total codebook size is set to be 512, TURN gets the highest cIoU and AUC.

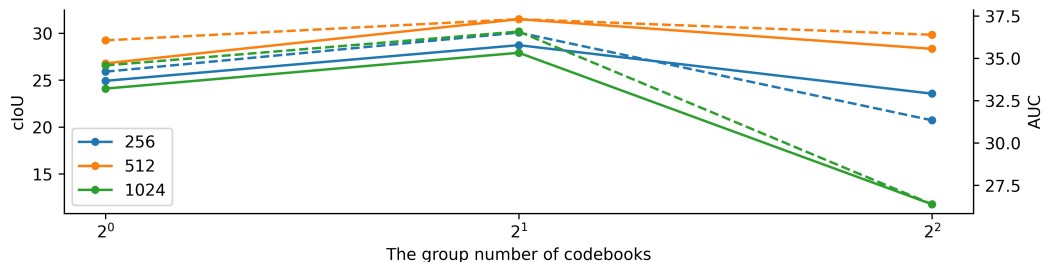

Figure 3: Effect of the hyper-parameters of codebooks on VGGSS. The curves of cIoU and AUC are drawn in the solid lines and the dashed lines, respectively.

**Choice of Reconstruction Settings** As formulated in Eq. 5, the reconstruction of text and audio data will be conducted in a *Self-Cross* manner, which is expected to improve the mutual information between text and audio pairs and eliminate modality and domain gaps between different data sources. To evaluate the effect of our reconstruction strategy, we further examine other combinations in the format of "*A-B*", where *A* denotes the reconstruction type adopted in the source domain and *B* denotes the one adopted in the intermediate domain. The performance comparison can be found in Figure 4. It's worth noting that "*No*" means "*no reconstruction will be applied.*", and "*Self / Cross*" means that "*the reconstruction will be conducted in a uni- / cross-modality manner*". From Figure 4, we can find that the reconstruction in each domain is essential and decisive to the final performance, and the *Self-Cross* one behaves superior to the other combinations.

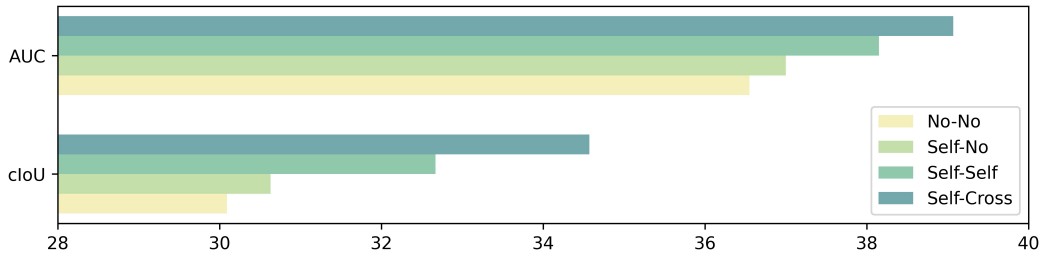

Figure 4: The performance of different reconstruction settings on VGGSS dataset.

## 4.4 Qualitative Analyses

In order to comprehensively evaluate the performance of our TURN, we will analyze some success cases on both the VGGSS and MUSIC-Solo datasets, which are shown in Figure 5. (The failure ones will be discussed in the supplementary material [5] .) From these cases, we can easily find that our

---

[5] And more details about experiments can also be found in the supplementary material.

model obtains the basic ability to discriminate the corresponding target according to the specified sound from a complex scene. Limited by the structural design, when faced with the multi-box scenarios of the VGGSS dataset, our model can only try to randomly choose one (c) or take their union (d) as the prediction among multiple equivalent candidates. Although this will result in a lower metric score, these cases can still qualitatively indicate the satisfactory performance of our proposed architecture.

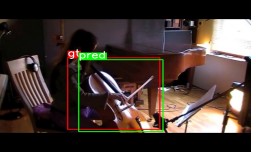 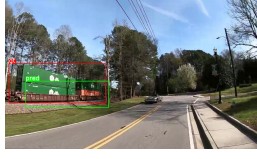 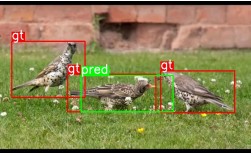 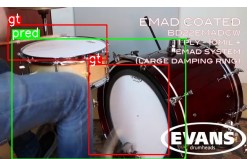

(a) Cello Performance     (b) Stack Train     (c) Thrush Birds     (d) Drum Set Performance

Figure 5: Success cases with one or more ground-truth annotations in MUSIC-Solo (a) and VGGSS (b, c, and d). The predictions and ground-truths are marked in green and red boxes, respectively.

## 5 Conclusion

In this paper, we develop TURN, an effective two-stream architecture for zero-resource sounding object localization, which consists of a localization stream and an alignment stream. We explore the task of sounding object localization from a zero-resource perspective for the first time and discuss the feasibility of transfer-based solutions in this setting. Without using any training data from the field of sounding object localization, our proposed TURN reaches a counterpart to other full-resource SOTA methods on publicly-available benchmarks (including MUSIC and VGGSS), which demonstrates the feasibility of our TURN. More detailed analyses further verify the effectiveness of the overall mechanism and architecture, and the qualitative analyses also provide a more intuitive and comprehensive perception of the results.

## Acknowledgments and Disclosure of Funding

This work is supported by Zhejiang Natural Science Foundation LR19F020006 and National Natural Science Foundation of China under Grant No.62072397 and No.61836002. And this research is also supported by Yiwise and Alibaba-Zhejiang University Joint Research Institute of Frontier Technologies.

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
