# Towards Effective Multi-Modal Interchanges in Zero-Resource Sounding Object Localization

**Yang Zhao**[*], **Chen Zhang**[*], **Haifeng Huang**[*], **Haoyuan Li, and Zhou Zhao** [†]
Zhejiang University
{awalk, zc99, huanghaifeng, lihaoyuan, zhaozhou}@zju.edu.cn

## A  Appendix

### A.1  Qualitative Study

#### A.1.1  Analysis on Failure Cases

In this section, we will further elaborate on the failure cases in VGGSS and MUSIC-Solo datasets. We first divide these cases into three categories for clarification and discuss them one by one as follows.

**Case 1: Inability to understand or identify the target object.**  Although the vast amount of training data covers lots of categories, objects and the corresponding sounds (such as *Lathe* and *Robot Vacuum* shown in Figure 1) that are rarely or not present in the training procedure may still appear during the testing phase. Our model may fail to figure out the target region without any extra information or knowledge about the characteristics or properties of these objects.

**Case 2: Inability to distinguish between multiple candidates of the same category.**  In some particular scenarios (such as the cases in Figure 2), multiple candidate objects of the same category might exist simultaneously, and it will be difficult to distinguish between them by some tiny imperceptible differences. It might be essential to consider more context information for accurate identification and discrimination when dealing with such cases.

**Case 3 & 4: Irreconcilable annotation gaps.**  As shown in Figure 3 and 4, it is straightforward to notice that our model tends to put the whole person into the predicted regions when faced with targets that are *part of* or *related to* people. This phenomenon may stem from the annotation bias that the bounding boxes related to people in image grounding datasets tend to contain the whole area of people, which makes the predicted results incorrectly larger in sounding object localization.

#### A.1.2  Analysis on Domain-Exclusivity

To illustrate the discriminant ability of the auxiliary classifier to *domain-exclusivity*, we sort all the data in each dataset according to the sampling weights and select the most and least domain-exclusive $m = 10$ ones. These items can be found in the *cases* folder of supplementary material. Looking into these samples, we can find that the auxiliary domain classifier generally achieves the expected function. Most of the data pairs with **lower** domain-exclusivity (i.e., **higher** sampling weight) describe visual entities that emit specific sounds, which can facilitate the process of knowledge transfer. And the items with **higher** domain-exclusivity (i.e., **lower** sampling weight) describe some *silent objects*, *ambient sound* or *noise*, which are actually redundant or obfuscating information to the other domain.

---

[*]Equal Contribution.
[†]Corresponding Author.

36th Conference on Neural Information Processing Systems (NeurIPS 2022).

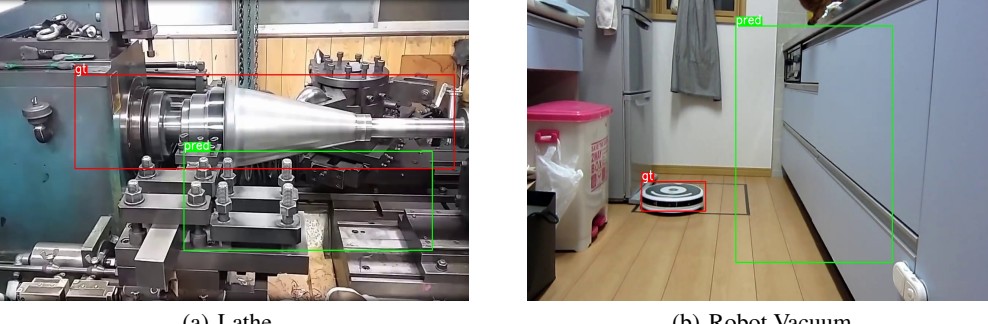

(a) Lathe                    (b) Robot Vacuum

Figure 1: **Failure Case 1** – Inability to understand or identify the target object.

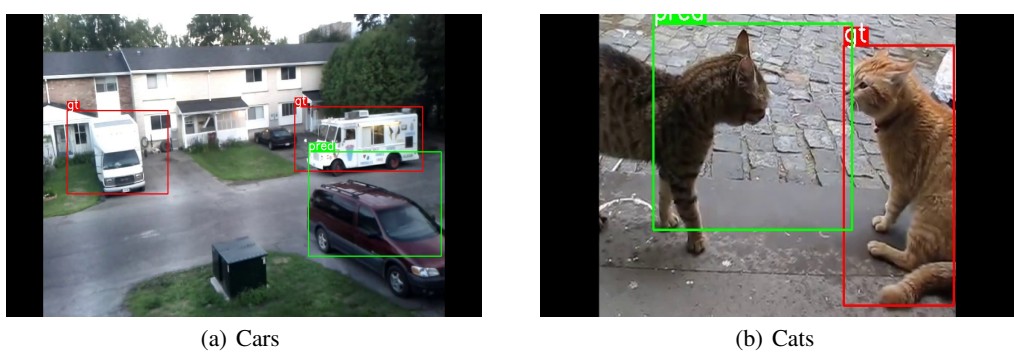

(a) Cars                    (b) Cats

Figure 2: **Failure Case 2** – Inability to distinguish between multiple candidates of the same category.

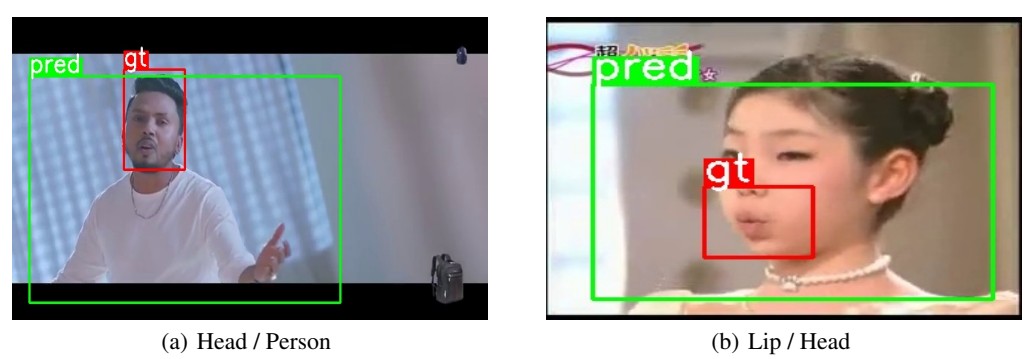

(a) Head / Person                    (b) Lip / Head

Figure 3: **Failure Case 3 (VGGSS)** – Inconsistency of scale / Irreconcilable annotation gaps.

## A.2 Implementation Details

### A.2.1 Inputs

Data of different modalities will be preprocessed as follows, regardless of their data domains.

- **Image:** The input size of image data is set as $H \times W = 384 \times 384$. In the training phase, we conduct the same data augmentation strategies as [10, 11, 3] where *random scale*, *random crop* and *random translate* are taken into practice. In the inference phase, the shortest edge will be stretched to $384$, and the entire image will be accordingly scaled.

- **Audio:** The audio waveforms will be converted into log-mel spectrograms with the sample rate of 22050 following [12] with a frame size of 2048 and hop size 1024, resulting in $L_m = 216$ and $c = 128$. For audio retrieval data used in training phase, *time masking* and

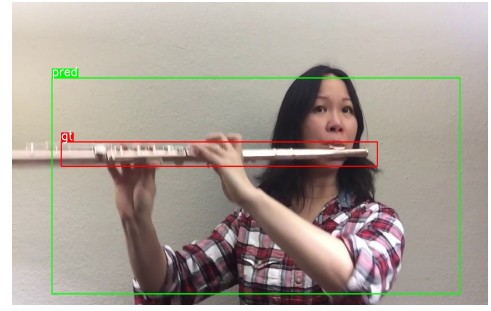
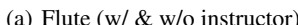
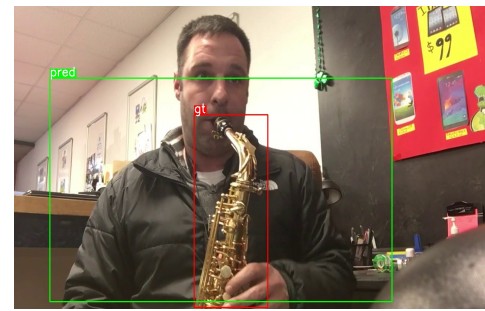

|  (a) Flute (w/ & w/o instructor) | (b) Saxophone (w/ & w/o instructor) |

Figure 4: **Failure Case 4 (MUSIC-Solo)** – Inconsistency of scale / Irreconcilable annotation gaps.

*frequency masking* of SpecAug [6] are applied to the corresponding log-mel spectrograms with the time masking length and stride as 64 and frequency masking length as 8 respectively.

- **Text:** All input words will be converted into lowercase and mapped as word indices via the tokenizer of pre-trained BERT [2] and the maximum length of text inputs $L_t$ is set to 30.

### A.2.2 Model Setting

Except for the parameters in the predefined structures, the dimension $d$ of intermediate representations and learnable parameters are set to be 256. In the *localization* stream, we utilize a pre-trained DETR encoder [1] with a R50 backbone to generate the object-aware embeddings $\hat{\mathbf{V}}$. The layer number, head number, and feed-forward dimension of transformer encoder $\Phi_v$ are set as 3, 4, and $4d$, respectively. As for the *alignment* stream, the length of log-mel spectrograms will be initially compressed as $L_a = 54$ by a stack of convolutional layers. The layer number, head number, and the feed-forward dimension of transformer encoders $\Phi_t$ and $\Phi_a$ are set to be 6, 8, and $4d$. The same configuration also works for the transformer decoders $\Omega_t$ and $\Omega_a$. Besides, the mask ratio in the reconstruction process increases linearly with the number of training epochs from 0.5 to 1. More details can be referred to in the uploaded code files.

### A.2.3 Training and Inference

In the training stage, the hyper-parameters controlling the weights of different regularization terms are set as $\lambda_1 = \lambda_2 = 1, \lambda_3 = 1, \lambda_4 = \lambda_5 = 0.1$ and $\beta = 0.25$. The temperature for contrastive loss $\tau_c$ is 0.1. And we employ AdamW optimizer proposed by Loshchilov and Hutter [5] with warm-up strategy [9] and cosine annealing learning [4], where the warm-up epoch is set as 1. The weight decay is set to be 1e-5, the maximal learning rate is set to be 1e-4, and the overall training process will last for 30 epochs. The mini-batch size of the source domain and intermediate domain will be balanced according to the scale of each corpus, and we fix the batch size of the source domain as $b_s = 64$. For the optimization of codebooks, we practically follow Van Den Oord et al. [8] and apply the exponential moving averages (EMA) mechanism to update the parameters in codebooks. As for the adaptive sampling strategy, the ratio $k$ is fixed at 5 for all the training data. The maximal learning rate is set as 1e-4 as well, but the training procedure will only last for two epochs. The training data for the auxiliary domain classifier are obtained by combining all the data from the source domain and intermediate domain. Moreover, although we use the subset of Flickr-SoundNet [7] as the dev set to select the best checkpoint of our model, it will also be all right to use the validation partition of source and intermediate domains, which can be more in line with the setting of zero-resource sounding object localization.