# OpenReview forum: "Towards Effective Multi-Modal Interchanges in Zero-Resource Sounding Object Localization"
_NeurIPS.cc/2022/Conference — NeurIPS 2022 Accept_

### Official Review · Reviewer_YYdX · 2022-06-27

**Rating:** 6
**Confidence:** 5
**Soundness:** 3 good
**Presentation:** 4 excellent
**Contribution:** 3 good

**Summary:**

This paper aims to locate sounding objects in an audiovisual scene without resorting to specialized training data for sound localization. It designs a new framework Two-stream Universal Referring localization Network to achieve multi-modal knowledge transfer among visual, text and audio modalities. Specifically, the localization stream locates objects from related texts using DETR structure, the alignment stream establishes a universal feature space shared by audio and text. In this way, it manages to localize the objects related to the specific sounds without extra annotations. The experiments show the effectiveness of the proposed method.

**Questions:**

My questions are stated in the weakness part, mostly on the experiments.

**Limitations:**

Yes, the authors solve the limitations of lacking benchmarks. And there is no obvious negative social impact.

**Strengths And Weaknesses:**

Strengths:
1. This paper is well motivated, and the idea of transferring knowledge from three modalities is interesting. It solves the sounding object localization task from a new perspective, i.e., directly regress bounding box instead of generate heatmaps, and does not require extra human annotations.
2. The paper is well written and easy to follow, and most technical details are well illustrated. Though some are borrowd from previous works, but generally it is a suitable application in this task and effective in multi-modal konwledge transfer.
3. The adaptive sampling strategy makes sense and the learning procedure of the sampling weight is interesting. It fully utilizes domain information and selects reasonable training samples for text audio knowledge transfer.
4. The results on single-source sounding object localization are promising. And the analysis on the failure cases is valuable. It demonstrates that this new formulation of sound localization instead of heatmaps is worth exploring and also figures out how to improve it.

Weaknesses:
1. The biggest concern is that this paper only performs experiments on single-source data, lacks the results on multi-source scenarios. On signle-source data, the sound localization is comparatively easy and the performance is mostly decided by the localization stream since the alignment is trivial. But on multi-source cases, it can validate whether the alignment stream establishes satisfactory semantic text audio alignment. And from my perspective, the image grounding task can help discriminate object semantics, and the text audio alignment also gather similar semantics, it should be able to deal with multi-source scenes with minor modifications. So I strongly suggest authors to try to expand the framework to more complex scenes for a test.
2. The visualizations in Sec.4.4 show that multi-box cases are diffcult to process. Deeper analysis on what causes this problem is desired. For example, when providing the text embedding as query how will the localization result be. Better to figure out whether it is a problem from localization stream or alignment stream.

---

> ### Author Response · Authors · 2022-07-30
> **Response to Reviewer YYdX**
>
> Thanks for your careful review and kindly comments on this paper.
>
> #### **[Weakness 1 & Weakness 2]**
> As shown in the last two rows of Table 1 and Table 2, we think the results have demonstrated that our alignment stream works well and plays a decisive role in the prediction. We notice that if a random query is given in the localization stream instead of the corresponding query, there will be a large performance drop in both datasets. It shows that the performance is not mostly decided by the localization stream. Besides, in the (b) case of Section 4.4, there also exist different kinds of vehicles in the image (a car and a stack train), but our model can still correctly identify the target object, which also demonstrates the performance of alignment stream in the qualitative aspect.
>
> We expand our method to more complex scenes according to your useful suggestions. More explanation and some extra experiments of multi-box and multi-sound scenarios can be found in the second point of "**About Evaluation Metrics & Multi-box / Multi-sound Scenarios**" in the official comments from the authors.

---

> > ### Comment · Reviewer_YYdX · 2022-08-09
> > **Thanks for your response**
> >
> > Thanks for your response. Given the experiment results on multi-sound cases, the proposed method cannot distinguish different semantics of sounding objects, but outputs the whole sounding area of mixed sounds. But since this paper adopts text modality, it should be feasible to design a series methods to decouple the query and return sounding objects of different semantics, which makes the model generalize to different scenarios. I hope authors can provide some analysis on this circumstance and look forward to further response.

---

> > > ### Author Response · Authors · 2022-08-09
> > > **Further Response to Reviewer YYdX**
> > >
> > > If individual boxes are needed for each sound component in the scenario of mixed sounds, the source domain should be designed as a one-to-many (one query & multiple targets) localization task, or extra sound separation modules will be needed in our architecture. Because from the aspect of task definition, both image grounding and sounding object localization aim to map a single query to a **single** target region. The scenario of mixed sounds is not actually a counterpart or a siamese task of image grounding.  And we think  the inability to generate a separate box for each object may not imply that our model cannot distinguish between different sounds, and we will continue to explore possible solutions to such issues in future work.

---

> ### Author Response · Authors · 2022-08-09
> **Looking forward to your reply**
>
> Thank you again for your thoughtful and constructive comments. If there are still concerns / open questions, we would be happy to hear and discuss them.

---

### Official Review · Reviewer_w7AN · 2022-07-10

**Rating:** 6
**Confidence:** 5
**Soundness:** 2 fair
**Presentation:** 2 fair
**Contribution:** 2 fair

**Summary:**

This paper studies sounding object localization from a zero-resource perspective. To solve this, the paper proposes a Two-stream Universal Referring localization Network(TURN), which includes a localization stream and an alignment stream. In order to solve the domain gap, an adaptive sampling strategy is used in this paper. There are many terms of loss function in this paper, the authors have carried out sufficient ablation experiments to prove the effectiveness of each part. The experiments in this paper prove the effectiveness of the model.

**Questions:**

1. In line 167，is each element in set $V$ $\hat{v}_i$ or $v_i$? From the following, should it be $v_i$?
2. Why does equation(5) use a cross-modality way in the intermediate domain? In the loss function, the contrastive learning loss has been used to align the audio modal and the text modal. What is the effect if the cross-modality way is not used here?
3. The result of DML and DSOL is heatmap. Is it fair to compare the IOU calculated by heatmap and the bounding box of ground truth with the one calculated by the bounding box and the bounding box of ground truth?

The authors are expected to solve the problems in the weakness and questions, then I can accordingly modify my ratings.

**Limitations:**

The authors have adequately addressed the limitations and potential negative societal impact of their work.

**Strengths And Weaknesses:**

### Strengths
+ This paper proposes a Two-stream Universal Referring localization Network(TURN) to combine two domains(Image Grounding and Audio Retrieval) to solve zero-resource sounding object localization.

+ An interesting point in this paper is that an additional classifier is trained to measure the domain gap of the sample, based on which the data is sampled for training.

+ This paper also conducted sufficient ablation experiments on each loss in the model, proving the effectiveness of each part. In addition, the authors analyze the failure cases in the appendix to illustrate the limitations of the method.

### Weakness

- In Line 37-Line 39, the paper mentioned that most of the current models generate a heatmap rather than a bounding box, which bring much inconvenience. However, the inconvenience is not described in detail in paper.

- Paper [1,2] also proposes a method to give the bounding box, which is not compared with the method in this paper.

- The paper does not explain how the texts of two domains are aligned during training. Although an adaptive sampling strategy is proposed to solve domain gap, in the training process, the texts from two domains may be irrelevant in the same batch.(For example, in the same training batch, the image grounding text description may be "a car was going on the road" but the text description in audio retrieval may be "a dog is barking on the road")

- The method in this paper can only give a single bounding box. For the mixed sound scenes, the method can not deal with it correctly.


[1]  D. Hu, et al., “Class-aware sounding objects localization via audiovisual correspondence,” IEEE Transactions on Pattern Analysis and Machine Intelligence, 2021.
[2] T. Afouras, et al. “Self-supervised object detection from audio-visual correspondence,” Proceedings of the IEEE/CVF Conference on Computer Vision and Pattern Recognition. 2022.

---

> ### Author Response · Authors · 2022-07-30
> **Response to Reviewer w7AN**
>
> Thanks for your detailed and instructive review comments. Here are the replies to your concerns.
> #### **[Weakness 1]**
> The inconveniences of heatmap-level prediction can be explained in multiple views.
> 1. From the application aspect, the box-level prediction has relatively small storage and transmission overhead. Besides, the box boundaries are easy to modify and edit, so users can revise the predictions or crop regions at a lower cost.
>
> 2. From the research aspect, the heatmap-based weakly-supervised methods usually suffer from instability in the training process. Besides, as a pair of siamese tasks, effective architectures or approaches in image grounding cannot get transplanted to the field of sounding object localization easily, which impedes the joint development of these two fields.
>
> #### **[Weakness 2]**
> [1] was the extension of [27] cited in our paper. The performance on Music-Solo is totally the same in [1] and [27], and [1] didn't report their performance on VGGSS.
>
> [2] did not report the performances in sounding object localization on the Music-Solo and VGGSS datasets. Moreover, this paper was published in CVPR2022 held in late June, which is actually concurrent to/after our NeurIPS submissions. We consider evaluating their methods on Music-Solo and VGGSS datasets, and adding the comparison after they release the codes.
>
> #### **[Weakness 3]**
> Sorry for making the misunderstanding. In the training process, we do not need to make explicit semantic alignment between texts from different sources. Specifically, the texts from image grounding will serve as queries in the localization stream, while the ones from audio retrieval will be aligned to the paired audios. We only need to construct a shared semantic space and eliminate the domain gap between them instead of aligning them in such a fine granularity.
>
> #### **[Weakness 4]**
> Please refer to the second point of "**About Evaluation Metrics & Multi-box / Multi-sound Scenarios**" in the official comments from the authors.
>
> #### **[Question 1]**
> Sorry for making confusion.  We will correct these annotations in the revision.
>
> #### **[Question 2]**
> We expect to improve the mutual information between text and audio data from the same pair and further eliminate modality differences through cross-modal reconstruction. We also conducted a detailed ablation study here, where we find that the cross-modality reconstruction indeed works better than other strategies.
> |  Setting |  cIoU@0.5 |    AUC    |
> |:--------:|:---------:|:---------:|
> | **(\*)** | **34.57** | **39.07** |
> |    (A)   |   30.09   |   36.55   |
> |   (B)   |   30.63   |   37.00   |
> |   (C)  |   32.67   |   38.15   |
>
> The source, intermediate, target domains in this analysis are Refcocog, Clotho and VGGSS, respectively.
>
> (A) No recon. in both source and intermediate domains.
>
> (B) Self recon. in the source domain; No recon. in the intermediate domain.
>
> (C) Self recon. in both source and intermediate domains.
>
> (\*): Self recon. in the source domain; Cross recon. in the intermediate domain.
>
> #### **[Question 3]**
> Please refer to the first point of "**About Evaluation Metrics & Multi-box / Multi-sound Scenarios**" in the official comments from the authors.
>
> [1] D. Hu, et al., “Class-aware sounding objects localization via audiovisual correspondence,” IEEE Transactions on Pattern Analysis and Machine Intelligence, 2021.
>
> [2] T. Afouras, et al. “Self-supervised object detection from audio-visual correspondence,” Proceedings of the IEEE/CVF Conference on Computer Vision and Pattern Recognition. 2022.

---

> ### Author Response · Authors · 2022-08-09
> **Looking forward to your reply**
>
> Thank you again for your thoughtful and constructive comments. If there are still concerns / open questions, we would be happy to hear and discuss them.

---

### Official Review · Reviewer_mRAH · 2022-07-12

**Rating:** 6
**Confidence:** 4
**Soundness:** 3 good
**Presentation:** 3 good
**Contribution:** 3 good

**Summary:**

* The paper proposes a method for training a sounding object localization model when there is no paired data available. It achieves this by transferring knowledge from paired data from overlapping modalities i.e. image-text and text-audio datasets.
* The idea is that by aligning the text representations from the different paired datasets into the same space, we can learn to match the accompanying audio representation with the appropriate visual embedding.
* The text and audio representations are projected into a common space using codebooks that are learned through the self-supervised training method of the vq-wav2vec [7] idea. The image representation is combined with the text representation from the codebooks and passed through a patch encoder to perform bounding box regression.
* The experimental results on the VGSS dataset shows that the proposed method can perform better than other methods that use supervised data for training.


**Questions:**

* Can you provide results from an experiment that combined annotated and un-annotated data?

**Limitations:**

The paper can include a discussion the biases of different data sources that can propagate to the results, especially when performing zero-shot recognition.

**Strengths And Weaknesses:**

### Strengths
* The proposed method builds on ideas developed for individual modalities and brings them together to learn multimodal models in low resource settings.
* The paper addresses the problem of sample weighting since there can be domain gaps in the source and intermediate domains. It uses an objective function that weighs the text embeddings such that they are hard to distinguish by a domain discriminator.
* The results on the VGSS dataset are significantly better than the methods that have been trained in a supervised manner.
* The paper also addresses the question of how to pick the source and intermediate domains via an ablation study.

### Weaknesses
* The IOU result on MUSIC-Solo is oddly very low compared to the other methods. The paper doesn’t provide any explanation or investigation for this result.
* The paper doesn’t include an experiment that shows how to combine the annotated data along with the un-annotated data. It would be interesting to see how much additional benefit can be obtained from using un-annotated data.

---

> ### Author Response · Authors · 2022-07-30
> **Response to Reviewer mRAH**
>
> Thanks for your valuable comments. The explanations for your concerns are shown below.
> #### **[Weakness 1]**
> The reasons for this phenomenon can be explained in the following aspects.
>
> 1. The annotation gap is difficult to eliminate. As shown in the failure case analysis of the appendix / supplementary materials, we can find that our model tends to regard the performers as a part of the sounding target, which stems from the annotation bias between the image grounding dataset and the sounding object localization dataset. So the union of the prediction box and the ground-truth one will be larger, and then the IoU metric will suffer a drop.
>
> 2. The music instruments have a lower proportion in the training set. The other models listed in Table 2 get trained and tested on the same domain of music instruments, but our model conducts a zero-resource generalization on the testing set. So the ratio of music instruments in the training dataset will affect the final result.
>
> 3. The performance of other SOTA works is not stable enough. According to the issue raised in the code repository https://github.com/DTaoo/Discriminative-Sounding-Objects-Localization/issues/10, some excellent performances may be hard to replicate steadily, and it seems that AUC is a more reliable metric.
> Besides, compared with all the other SOTA works, we think our model has actually already achieved an acceptable IoU metric. And we will add the analysis above in the next version.
>
> #### **[Weakness 2 & Question 1]**
> We also agree that it would be interesting to use the un-annotated training data of the sounding object localization dataset and evaluate the improvement of introducing the target domain. However, it needs us to design and add some additional components to the existing architecture so as to utilize the un-annotated data properly. We think it will become a semi-supervised task based on some adaptation mechanism, which is a little bit inconsistent with our main topic of "zero-resource sounding object localization". And we will discuss this topic and conduct related experiments in future work.

---

> > ### Comment · Reviewer_mRAH · 2022-08-08
> > **Rebuttal acknowledgement**
> >
> > Thanks authors for the response. I was mostly interested in understanding the reason for some odd results in the experiments. I see that there are a couple of data related effects that cause that result. Please try to incorporate this rebuttal in a brief format in the paper. I hope that explicitly calling out these characteristics will prevent a future reader to think something might be fishy in the results.

---

> > > ### Author Response · Authors · 2022-08-08
> > > **Response to Reviewer mRAH**
> > >
> > > Thank you again for your suggestion! We will try to include this rebuttal in a short form in the next version of this paper to help the reader gain a better understanding.

---

### Official Review · Reviewer_oJxK · 2022-07-12

**Rating:** 4
**Confidence:** 4
**Soundness:** 2 fair
**Presentation:** 3 good
**Contribution:** 2 fair

**Summary:**

In order to localize the object emitting a specified sound, i.e., sounding object localization, this paper proposes a Two-stream Universal Referring localization Network (TURN) concerning the localization stream and alignment stream. The TURN model correlates the referring image localization via the localization stream and text-audio semantic space via the alignment stream. Such a mechanism transfers the audio query through the learned codebooks, retrieved from the text-audio semantic space, to indirectly trigger the referring object detector. In addition, the authors provide an adaptive sampling strategy for helping training. The experiments show the proposed model on par with the existing methods.

**Questions:**

The major concern is the motivation to treat the sounding object localization task as a combination of image grounding and audio retrieval tasks. Such treatment seems complicated to the target task and the resulting model. In particular, previous methods [48, A] showed that the heatmaps retrieved from the sounding object localization could generate the corresponding boxes.

**Limitations:**

No limitations and potential negative societal impact included.

**Strengths And Weaknesses:**

[Strengths]
+ The idea of utilizing the zero-resource perspective for tackling sounding object localization is interesting.
+ The manuscript is well organized and has adequate references.

[Weaknesses]
- The technical novelty is limited since the TURN model is mainly composed of DETR [10], VQ-VAE [55], and VQ-WAV2VEC [7]. The TURN model shows an application employing these methods [7,10,55] with an additional adaptive sampling strategy for helping the model training.
- The proposed method makes the task of sounding object localization from the audio-image matching problem to the audio-text-image matching one. The proposed method seemingly complicated the original task of acquiring the box-level prediction. The proposed method also implies that the extra data are needed yet ignoring the available training data from the field of sounding object localization. However, the performance gain is not significant.
- While evaluating the model, does the TURN model evaluated with the box-level prediction? It seems that the box-level prediction is not the same as the heat-map prediction, as shown in [A].
- Figure 2 needs more descriptions and notations to gain a better understanding.
- What is the so-called `patch’ in line 168 used in a DETR backbone [10]?
- The loss terms mentioned in section 3.5 should be consistent with the notations in figure 2.
- The reference is not sufficient. For example, some related methods are shown as follows.
[A] Arda Senocak, Hyeonggon Ryu, Junsik Kim, In So Kweon: Less Can Be More: Sound Source Localization With a Classification Model. WACV 2022: 577-586.

---

> ### Author Response · Authors · 2022-07-30
> **Response to Reviewer oJxK**
>
> Thanks for your kind suggestions. The replies to your concerns are listed as follows.
> #### **[Weakness 1]**
> In this paper, our purpose focuses more on exploring a possible transfer learning paradigm between siamese tasks and designing a universal framework in the field of referring vision comprehension, rather than merely devising an application to generate box-level predictions or improve performance in the field of sounding object localization. And the modules from DETR, VQ-VAE, and VQ-WAV2VEC are just bricks to help us build the architecture of our proposed approach, which can be replaced by other off-the-shelf modules with similar functions.
>
> #### **[Weakness 2 & Question 1]**
> For generating box-level predictions, [48, A] actually use a multi-stage strategy, i.e., generate heatmaps first and then transform them into the corresponding boxes through hand-crafted rules heuristically. This strategy is independent of the proposed architecture so that any heatmap-based approaches can be accordingly converted into box-based ones. But such a prediction cannot be learned or applied end-to-end and requires extra knowledge to design heuristic rules and choose thresholds. In contrast, our model can make straight inferences without reliance on handmade transformation rules.
>
> As for data usage, we introduce the extra datasets and do not use the original training samples because the lack of annotations will lead to a relatively unprecise and ineffective alignment, thus impairing the learning efficiency of models. And actually, the total amount of extra data we used is less than that of the target training data, and we did not fine-tune our model to fit different test sets. Under this circumstance, our model can achieve comparable performance with other SOTA methods in the *unseen* domain, which has shown the feasibility of our proposed approach. Besides, the challenging zero-resource setting can best reflect the generalization performance in such a cross-task transfer learning scenario, which is another reason we do not use the original training data.
>
> #### **[Weakness 3]**
> Please refer to the first point of "**About Evaluation Metrics & Multi-box / Multi-sound Scenarios**" in the official comments from the authors.
>
> #### **[Weakness 4 & 6 & 7]**
> Thanks for your advice. We will carefully check the annotations and add more descriptions of Figure 2 and references to other related works in the revision.
> #### **[Weakness 5]**
> Sorry for causing the confusion. The annotation $n_v$ is equivalent to $H \times W$ introduced in Section 3.2 of [10], representing the input sequence length of the DETR encoder.

---

> > ### Comment · Reviewer_oJxK · 2022-08-08
> > **Thanks for the authors' response.**
> >
> > Thanks for the authors' response. Treating the target task by integrating two related tasks is more complicated than directly designing a model tailored for that target task. Though it is known that the adopted modules from DETR, VQ-VAE, and VQ-WAV2VEC are replaceable bricks to build the proposed architecture. However, the model performance of the target task is also restricted by the modules derived from the two related tasks, right? How to assess the architecture generalization? Is it possible to apply such architecture to the other tasks?

---

> > > ### Author Response · Authors · 2022-08-08
> > > **Further Response to Reviewer oJxK**
> > >
> > > Thank you for your reply.
> > > First, the model performance of the target task is indeed restricted by the modules derived from the two related tasks. However, only the tasks equipped with sufficient training data will be chosen as the related task in this prototype. Therefore, most of the selectable related modules tend to perform well enough to cover our target scenario.
> > >
> > > Moreover, from an overall perspective, our architecture can function as the bridge of transferring knowledge and information in various scenarios instead of only focusing on the performance of a single task with specially designed model architecture, and it can also be generalized to other understanding or generation tasks.
> > >
> > > For example, the amount of data in the audio caption domain is still relatively limited. We can use the naturally aligned video-audio pairs and the data from the field of video caption to help us obtain more knowledge in the audio caption. Besides, in the field of semantic-guided image generation and manipulation, sounds[1] and texts[2] can both be regarded as guidance information, in which we can also devise a modality-independent method to help transfer knowledge or implement a universal architecture.
> > >
> > > By choosing the field of sounding object localization as the starting point, we not only present a possible solution to the problem of inaccurate annotations in this field but also demonstrate the good generalization ability of our proposed strategy through clear quantitative comparison and intuitive qualitative analysis.
> > >
> > > [1] Lee, Seung Hyun et al. “Sound-Guided Semantic Image Manipulation.” ArXiv abs/2112.00007 (2021): n. pag.
> > >
> > > [2] Patashnik, Or et al. “StyleCLIP: Text-Driven Manipulation of StyleGAN Imagery.” 2021 IEEE/CVF International Conference on Computer Vision (ICCV) (2021): 2065-2074.

---

### Author Response · Authors · 2022-07-30
**About Evaluation Metrics & Multi-box / Multi-sound Scenarios**

Several reviewers have raised some common issues, so we will make a response to these questions together here.
### i) Evaluation Metrics
Almost all the works in the field of sounding object localization use cIoU (consensus Intersection over Union) based on *heatmap* as their evaluation metrics. For a fair comparison, we first mapped the predicted box into a heatmap, in which the pixels within the box are set to 1, and the others are set to 0 in the testing phase, and then computed cIoU on the generated heatmap so as to keep consistent with other methods.

### ii) Multi-box / Multi-sound Scenarios
1. **Multi-box**: To clearly distinguish from **multi-sound** scenes, we define **multi-box** scenarios as there are multiple objects in the given picture that emit the same/similar sound. As illustrated in Section 4.4, constrained by the structure of the **localization stream**, our model can only randomly choose one or take their union as the prediction among multiple equivalent candidates, and it actually makes a relatively ideal prediction under this situation, which we think has already demonstrated the feasibility of our proposed transfer learning mechanism. We will discuss and explore structures of *localization stream* that can generate multiple bounding boxes in future works.

2. **Multi-sound**: In the **multi-sound** scenarios, multiple objects will emit different kinds of sounds simultaneously, which will get mixed into the given query. While for the other works in sounding object localization, mixed sounds' processing (if there is) always relies on an extra module or mechanism to help the model separate different sounds or targets[1, 2]. We conducted experiments on the mixed-sound datasets, including Music-Duet and DailyLife [1], and the results are listed below. It is worth mentioning that [1] conducts a class-aware IoU on these datasets, but our model cannot generate an extra class label as a reference. Therefore the evaluation will be different from [1], i.e., we will regard every annotated box as a part of the ground-truth for the mixed sound and calculate the consensus IoU result. And we will discuss possible solutions to deal with multi-sound scenarios in a finer granularity in future work.
|   Dataset  |      Setting | cIoU@0.5 |  AUC  |
|:----------:|:----------------:|:--------:|:-----:|
| MUSIC-Duet | w/o Query |   5.44   | 22.56 |
| MUSIC-Duet | w/ Query  |   16.06  | 33.77 |
|  DailyLife | w/o Query |   8.54   | 27.68 |
|  DailyLife | w/ Query  |   31.71  | 42.13 |

The source and intermediate domains in these experiments are Refcocog and Clotho, respectively.

[1] D. Hu, et al., “Class-aware sounding objects localization via audiovisual correspondence,” IEEE Transactions on Pattern Analysis and Machine Intelligence, 2021.

[2] Hang Zhao, Chuang Gan, Andrew Rouditchenko, Carl Vondrick, Josh H. McDermott, and Antonio Torralba. The sound of pixels. ArXiv, abs/1804.03160, 2018.

---

### Author Response · Authors · 2022-08-08
**We're sincerely looking forward to your reply and further discuss.**

Dear reviewers,

We really appreciate your insightful suggestions and valuable comments. In the previously uploaded replies, we have tried our best to address your questions one by one and supplemented more experiment results.
We sincerely look forward to your further discussions and comments on our response. And we are open to any discussion to improve our paper.

Best Wishes! :)

The authors.

---

### Meta-Review · Area_Chair_eTX6 · 2022-08-23

**Recommendation:** Accept
**Confidence:** Certain

**Metareview:**

This paper presents an effective transfer-based two-stream architecture for zero-resource sounding object localization, which is an interesting paper that will benefit research in zero-resource tasks in multimodal settings. The experiment results are pretty impressive, both quantitatively and qualitatively. The rebuttal successfully addressed some of the major concerns and, in the end, there is a general consensus about accepting the paper.

**Award:**

No

---

### Decision · Program_Chairs · 2022-09-14

Accept